# Early Gonadal Development and Sex Determination in Mammal

**DOI:** 10.3390/ijms23147500

**Published:** 2022-07-06

**Authors:** Yanshe Xie, Changhua Wu, Zicong Li, Zhenfang Wu, Linjun Hong

**Affiliations:** 1National Engineering Research Center for Breeding Swine Industry, College of Animal Science, South China Agricultural University, Guangzhou 510630, China; xys@stu.scau.edu.cn (Y.X.); wuchanghua@stu.scau.edu.cn (C.W.); lizicong@scau.edu.cn (Z.L.); 2Guangdong Provincial Key Laboratory of Agro-Animal Genomics and Molecular Breeding, South China Agricultural University, Guangzhou 510630, China

**Keywords:** gonadal development, sex determination, primordial germ cell, genes

## Abstract

Sex determination is crucial for the transmission of genetic information through generations. In mammal, this process is primarily regulated by an antagonistic network of sex-related genes beginning in embryonic development and continuing throughout life. Nonetheless, abnormal expression of these sex-related genes will lead to reproductive organ and germline abnormalities, resulting in disorders of sex development (DSD) and infertility. On the other hand, it is possible to predetermine the sex of animal offspring by artificially regulating sex-related gene expression, a recent research hotspot. In this paper, we reviewed recent research that has improved our understanding of the mechanisms underlying the development of the gonad and primordial germ cells (PGCs), progenitors of the germline, to provide new directions for the treatment of DSD and infertility, both of which involve manipulating the sex ratio of livestock offspring.

## 1. Introduction

The most important issues in human reproductive medicine are disorders of sex development (DSD) and infertility. According to data, the proportion of DSD patients ranges between 1:2000–1:4500 [1]; however, the underlying mechanisms of this congenital disease remain unclear, and genetic diagnostic cannot be performed on up to 75% of patients [2]. As for infertility, 8–12% of couples reproductive-aged are affected by this condition globally [3], which can also be caused by genetic background. In order to diagnose and treat DSD and infertility, a substantial amount of research has been devoted to sex determination, whose abnormality can result in these. In mammal, this process is governed by sex chromosomes [4] and involves the sexual differentiation of bipotential gonads and primordial germ cells (PGCs) [5]. In livestock production, the molecular mechanism of sex determination has been a research hotspot for many years because it has the potential to significantly increase production efficiency; however, a lack of theoretical knowledge prevents commercial application impossible [6]. In this review, we focused primarily on molecular, cellular, and genetic studies on early gonadal development and sexual differentiation of bipotential gonads, both with PGC formation, migration, and gender-specific differentiation. Although some studies in this field have been reviewed elsewhere [7,8,9,10,11,12], we summarized these review articles and combined single-cell RNA sequencing results on the basis of the original research to discuss some potential mechanistic links between bipotential gonads and PGC and find genes that play key roles in multiple stages of early gonadal development and sex determination. On the other hand, we described novel gene editing experiments that lead to sex reversal, potential gender differences before sex-specific differentiation, and the function of epigenetic regulation in these processes, with the hope of informing future studies on abnormal development of reproductive organs and infertility in humans, as well as sex manipulation technologies in livestock.

## 2. Genital Ridge Formation

In mammal, both the testis and ovary develop from the genital ridge (GR), which first appears at approximately four-five weeks of pregnancy in humans [13] and around embryonic day (E) nine and a half in mice [8], when coelomic epithelial cells begin to proliferate on the ventromedial surface of the mesonephros [14,15,16]. Each mesonephros contains a Wolffian duct and a Müllerian duct, which give rise to the epididymis, vas deferens, and seminal vesicles [17,18] or the fallopian tubes, uterus, and a portion of the vagina [19]. At around 32 days of pregnancy in humans (E10.5 in mice), coelomic epithelial cells differentiate into two distinct somatic precursor lineages (supporting cell precursors and steroidogenic cell precursors) [20,21]. Recent single-cell sequencing research has confirmed that mammalian gonadal cells originate from the same progenitor [22,23]. However, little is known about the development of the gonad prior to sex determination, as the bipotential gonad contains multiple uncharacterized subpopulations lacking specific markers [22]. In the last few decades, knockout mice models and mutation data from DSD patients have been used extensively to investigate transcription factors essential for genital ridge formation. These important genes are detailed in Table 1 and Figure 1. However, these genes, such as *GATA4*, *POD1*, *PBX1*, and *ODD1*, were also involved in the formation of various organs [24], which hinders the effectiveness of research in the field.

## 3. Differentiation of the Bipotential Gonads

At approximately 6–7 weeks of pregnancy in humans (E12.5 in mice), testis cords are observed in XY gonads, indicating the start of sex differentiation in bipotential gonads [51,52,53,54]. Controlled by gene expression dynamics and antagonistic genetic programs, sexual differentiation begins when the antagonistic network’s balance is tilted toward males or females. Furthermore, these antagonistic genetic programs will be maintained into adulthood to preserve gonad stability and reproductive capacity [55]. The *SRY* on the Y chromosome is the “master switch” for testis determination in mammal. When *SRY* is expressed in bipotential gonads during a critical window of fetal development, *SOX9* expression and a male-promoting regulatory network are activated, resulting in testis differentiation. In contrast, ovary differentiation will be triggered when the balance is tilted towards a female-promoting regulatory network [56]. However, recent research has not identified genes with a similar function role to *SRY* in female sexual differentiation. By inhibiting *SOX9* expression, *WNT4*, *RSPO1*, and *FOXL2* were previously involved in the ovarian-determination pathway.

### 3.1. Sex-Determining Region Y, SRY, and Sox9

In 1989, a 35-kb region from the Y chromosome was identified in 46,XX DSD patients, and it was believed to be the possible carrier of the testis-determining factor gene [57]. A year later, an open reading frame (ORF) encoding a new gene, later designated *SRY* (sex-determining region Y), was discovered in this Y fragment [58]. The *SRY* encodes a transcription factor with an N-terminal domain (NTD), a high conserved mobility group (HMG) domain, and a C-terminal domain (CTD) [59,60]. The *SRY* mutation analysis revealed that the NTD in *SRY* is associated with nuclear importation [61,62], whereas the CTD may contribute to the conformation and function of *SRY* [63] and be required for *SOX9* activation [60]. Most human male-to-female sex reversal syndrome cases are believed to be caused by a mutation located in the HMG domain of *SRY* [64,65]. Recent gene-editing research demonstrated that knockout of the HMG domain of the porcine *SRY* gene could result in male-to-female sex reversal [66]; however, additional tests are necessary to confirm the fertility of these transgenic animals. Interestingly, a study found that XX mice carrying a chimeric *SRY*/*SOX* construct (replacing the HMG domain of *SRY* with the HMG domain of *SOX3* or *SOX9*) also exhibited sexual reversal [67], suggesting that *SOX3* and *SOX9* HMG domains can functionally replace *SRY* HMG domain. In addition, a two-exon *SRY* transcript was required for male testis determination, disproving the theory that *SRY* is a single-exon gene [68].

During gonadal development in mice, the *SRY* is initially expressed in Sertoli cell progenitors at E10.5, reaches its maximum expression at E11.5, and then disappears at E12.5. In contrast, *SRY* expression in humans begins around 41 days of pregnancy and peaks at 44 days [69]. In contrast to mice, human *SRY* expression gradually decreases to a base level around 60 days of pregnancy and is maintained until adulthood [70]. In addition, *SRY* expression in the post-testis determination stage has been detected in goats, sheep, pigs, rabbits, and cattle [71], prompting additional research into the mechanism of the male-promoting regulatory network in mice following sex determination. *SRY* expression is non-synchronous in the gonad; the wave of *SRY* expression moves from the center to the poles between E10.5 and E12.5, limiting transcriptomics research on sex determination to some extent [72,73].

The *SOX9* expression reaches a plateau around 48 days of pregnancy in human testis [70] due to the synergistic action of *SRY* and *NR5A1* [74] (E11.5–12.5 in mice [75]), thereby activating the male-promoting regulatory network and testis determination [76,77], followed by suppressing *SRY* expression and binding to *NR5A1* to maintain expression [78]. In addition, *FGF9* and *PGD2* signaling pathways are activated after activation of testis-related genes, repression of anti-testis genes, inducing Supporting-to-Sertoli cell differentiation [79]. The *SOX9* was sufficient to induce testis differentiation in the absence of *SRY* in transgenic XX gonads [80,81], which was later confirmed in a 46,XX mosaic male patient [82]. Therefore, *SRY* may only regulate *SOX9* expression during testis development, which requires further investigation.

### 3.2. WNT Family Member 4 (WNT4)

The *WNT4* is a member of the *WNT* family, essential for early embryonic development, the transition between naive and primed embryonic stem cells (ESC), and tissue homeostasis in adults [83]. The *WNT4* is initially expressed in the undifferentiated early gonad at E11.25 [84], and *WNT4* knockout translated into a significant increase in steroidogenic cells in both sexes [85]. In addition, the proliferation of coelomic epithelial cells was reported to be inhibited in the early gonads of *WNT4*-*RSPO1*-double-knockout mice [84], indicating that *WNT4* may play the same role in both sexes during the early stages of gonadal development. Moreover, Müllerian duct formation failed in male and female *WNT4*^−/−^ mice [86]. Similarly, *WNT4* mutant 46,XX was found to have Müllerian duct abnormalities [87] and regression [88]. This suggests that *WNT4* is necessary for regulating the histogeny of the Müllerian duct in both sexes.

The *WNT4* is a component of the *WNT*/*β-catenin* signaling pathway and is essential in female sexual differentiation. The activation of the *WNT*/*β-catenin* signaling pathway is first detected in both sexes’ bipotential gonads at E11.5, acting as an anti-testicular agent by limiting the expression of *SOX9* [84], but is downregulated by *SRY* in males [89]. An increase in *WNT4* copies in humans was shown to result in a male-to-female sex reversal in 46,XY patients [90], while *WNT4* inactivation or mutation resulted in sex reversion–kidneys, adrenal, and lung dysgenesis (SERKAL) syndrome [91] or virilization [88]. The *WNT4* participates in accessory gland development by regulating hormone secretion, e.g., in *WNT4*^−/−^ XX mice, genes involved in testosterone (a hormone involved in the formation of the epididymis, vas deferens, and seminal vesicles) synthesis were found elevated [92]. In contrast, steroidogenic enzymes 3β-hydroxysteroid dehydrogenase and 17α-hydroxylase, which are required for testosterone synthesis, were expressed in ovaries [86]. However, these transgenic models do not affect steroidogenic cell differentiation [93], revealing that *WNT4* inhibits testosterone secretion by antagonizing steroidogenic cell migration rather than steroidogenic cell differentiation [94]. In humans, *WNT4* mutant 46,XX patients were reported to suffer hyperandrogenism [87].

The *WNT4* is required after sexual differentiation to prevent the formation of testis-specific vasculature, one of the earliest morphological changes during testicular differentiation [93,95]. In addition, it plays an essential role in the survival of oocytes and the maintenance of ovarian function [93,96]. Moreover, it is required for secreting steroid hormones in granulosa cells, which regulate normal ovarian follicle development and female fertility [97].

### 3.3. R-Spondin 1 (RSPO1)

The *RSPO1* was discovered in the dorsal neural tube of mice in 2004 [98]. Since then, the *RSPO1* family has been extensively studied, and the other three family members (*RSPO2*, *RSPO3*, and *RSPO4*) were discovered later. In mammal, these four *RSPO1* family members have similar domain organization and are essential for embryogenesis, development, and tumorigenesis [98,99,100].

The *RSPO1* is mainly expressed in mice XX gonadal somatic cells during ovary determination and suppressed in mice XY gonad, with only interstitial cells having low expression [101,102]. Loss-of-function experiment showed that *RSPO1* knockout led to sex reversal and formation of ovotestis in XX mice [103]. In humans, the *RSPO1* mutation caused hermaphroditism [104], palmoplantar hyperkeratosis, and squamous cell carcinoma [101]. Furthermore, RSPO1 functions conservatively in various vertebrates during ovarian development. In a recent study, goat BAC clones containing the *RSPO1* gene (*gRSPO1*) were injected into mouse oocytes, which resulted in the restoration of sex-reversal in *RSPO1* knockout XX mice [105]. Although the function of *RSPO1* in inhibiting testicular differentiation still requires further research, the result described above provides essential insights into DSD treatment.

With the deepening of research, RSPO1 protein has been identified as an agonist of the *WNT*/*β-catenin* signaling pathway [106]; *RSPO1*^−/−^ mice demonstrated an absence of activation of *WNT4* [102]. In addition, the ovarian phenotype of *RSPO1* knockout mice recapitulated with those of *WNT4* knockout female mice [107]. The *RSPO1* mutation 46,XX ovotestis, reduced expression of β-catenin protein and *WNT4* mRNA, restricted ovarian differentiation. Transfection of *RSPO1* resulted in activation of the β-catenin responsive TOPFLASH reporter (1.8-fold maximum), whereas *RSPO1* and *CTNNB1* (encoding β-catenin) synergy resulted in a 10-folds increased activation [108]. Above all, *RSPO1* functions as an enhancer of β-catenin signaling during early ovary development. Interestingly, a novel role of *RSPO1* in steroid hormone secretion independent of WNT/β-catenin signaling was discovered. After luminal cells-specific *RSPO1* knockout, *ESR1* (estrogen receptor alpha) expression was decreased, and mammary side branches were reduced. However, *ESR1* expression was increased after luminal cell-specific knockout of *WNT4*, both with the attenuation of WNT/β-catenin signaling activities [109], revealing *RSPO1* may involve in other signaling pathways that regulate female sexual differentiation.

The *RSPO1* is also reported to participate in oocyte differentiation and meiosis after sex determination, as germ cell proliferation, *STRA8* (early meiotic marker) expression, and the number of germ cells entering meiosis were all reported impaired in the *RSPO1*^−/−^ fetal ovary [110]. However, in human disease, *RSPO1* was found to promote progression in ovarian cancer by increasing the proliferation and migration of ovarian cancer cells and reducing ovarian cancer cells’ apoptosis [111].

### 3.4. Forkhead Box L2 (FOXL2)

The *FOXL2* is one of the earliest markers of ovary differentiation in a mammal, which is sexual-specific and expressed in female gonads after E12.5 [112]. In *FOXL2*-knockout XX mice, granulosa cells and steroidogenic theca cells were reprogrammed into Sertoli-like cells and Leydig-like cells under the repression of *SOX9* [113]. In the in vitro system, up-expression of *NR5A1* was antagonized by *FOXL2*, and a 2-fold increase in *NR5A1* expression was detected in *FOXL2*^−/−^ mice relative to wild-type mutant [114]. According to findings, *FOXL2* may regulate early ovarian development by directly suppressing the expression of testis-specific genes. Although *FOXL2* plays a vital role in ovarian development in goats, it is more involved in fetal development than postnatal maintenance when compared to mice [115]. In XY transgenic mice, over-expression of *FOXL2* led to the impairment of testis tubule differentiation [116], and *RSPO1*-*FOXL2*-double-knockout mice showed a similar phenotype earlier stage of sex reversal than *RSPO1* knockout mice, revealing a potential interaction between these two female sex determination genes [117]. 

The *FOXL2* becomes involved in follicle development by inducing Follicle-stimulating hormone (FSH) synthesis following sex determination [118,119], whose expression is regulated by ovarian hormones [120,121]. Furthermore, *FOXL2* plays a role in the development and maintenance of the ovary via interacting with *STAR* [122], *ESR2* [123]), and *p27* [124]. Moreover, *FOXL2* is expressed in the other components of the female reproductive tract, including the uterus, cervix, and oviduct, and plays a crucial role in postnatal uterine maturation [125]. The *FOXL2* mutations are linked to Blepharophimosis-Ptosis-Epicanthus Inversus syndrome (BPES) [126,127,128], adult ovarian granulosa-cell tumor [129], testicular adult-type granulosa cell tumors [130,131], ovarian Sertoli-Leydig cell tumors [132], incompletely differentiated sex cord-stromal tumors [131] and ovarian sex cord-stromal tumors [133] in human.

When these sexual differentiation-related genes are taken together, they regulate the testis- and ovarian-determination network (listed in Table 2 and Figure 1) during embryonic development and throughout adulthood. They are also associated with developing other organs, reproductive capacity, and health. As a result, research aimed at developing animal models and modifying offspring sex ratios using gene-editing technology has been hampered for a long time due to organ failure. Although some research has investigated the underlying mechanisms of these sexual differentiation-related genes and has produced sexual reversal offspring, the development of offspring reproductive organs was significantly retarded. The number of available knockout offspring is lacking, limiting the study on growth performance. In addition, recent research has emphasized the importance of epigenetics in regulating sexual differentiation [134], inspiring future research in exploring the function of DNA methylation, histone modifications, non-coding RNA, and RNA methylation during sex determination gonads. 

## 4. Current Knowledge of PGCs

The PGCs are distinct stem cells that can give rise to other stem cell types and pass on their genome to the next generation. PGC research offers new hope for treating infertility patients by in vitro mediating PGC differentiation, even though germ cell yields remain low. As a result, research on the formation of PGCs will be an important future research direction envisaged to promote in vitro derivation of human PGCs. Furthermore, there may be gender differences during the migration and differentiation of PGCs, which could provide a theoretical foundation for manipulating offspring sex ratios in livestock production by changing the ratio of Y- and X-chromosome-bearing sperm through gene editing.

### 4.1. Formation of PGCs

PGCs originate from a subpopulation of cells in the proximal epiblast (PE) at around two weeks of pregnancy in humans (around E6.5 in mice) [157]. Subsequently, these cells cluster and are located in the base of incipient allantois [158]. Current research has identified that bone morphogenic proteins (BMPs) mainly induce PGCs specification signals secreted from surrounding extraembryonic ectoderm (*BMP4*, *BMP8b*) [159,160] and visceral endoderm (*BMP2*) [161]. However, BMPs signals alone could not determine PGCs fate because only a subset of PE cells can induce differentiation into germ cells. Therefore, several in vivo and in vitro studies identified positive and negative signals directing PGC fate (listed in Table 3 and Figure 1). Furthermore, significant differences in PGC formation regulatory actions have been observed between humans and mice, such as *SOX2* is required for PGC development in mice, while *SOX17* is required in humans [162,163,164]. Moreover, *KLF4* is only involved in maintaining pluripotency in human PGCs [165]. Interestingly, LncPGCAT-1 was found to positively regulate the formation of PGCs by elevating the expression of *Cvh* and *C-kit* and repressing the *NANOG* in vitro and in vivo [166], providing a new direction for research into the underlying biology of PGCs formation. In addition, recent single-cell sequencing research showed that the germline development between bovines and humans [167] and between mice and humans [165] were similar, which may provide new model organisms for the research on the development of PGCs.

### 4.2. In Vitro Derivation of PGCs

To examine the mammalian germline’s developmental mechanism, mice PGCs were first isolated in 1982 [184]. Since then, PGCs of other species have been successfully isolated, including goats [185], rabbits [186], sheep [187], and humans [188]. However, due to the low PGCs generation rate, the current research was devoted to deriving PGCs from pluripotent cells. So far, primordial-germ-cell-like cells (PGCLCs) [189] and long-term expanded PGCLCs [190] have been developed to generate fertile mice oocytes [191] and produce offspring [192] in vitro. On the other hand, the same gene expression patterns were observed for human PGCLCs and PGCs [165], cementing the feasibility of researching PGCs formation in vitro. Several genes important in PGCs formation and maintenance, such as *TFAP2C* [193,194], *SSEA1* [195], *DND1* [196], and *SOX15* [197], were identified by employing sequencing technology, cell biology techniques, and genome editing technology on PGCLCs. However, there were still many challenges ahead; for instance, human PGCLCs derived in vitro could not meiosis completely during the embryonic stage [198]. Therefore, recent research is dedicated to the optimization of PGCs derivation routes.

According to recent studies, niche environments are important for differentiating human PGCs from pluripotent cells. Franklin D. West et al. discovered that co-culturing with mouse embryonic fibroblasts increased the expression of germ-cell-specific genes [199]. One year later, human fetal gonadal stromal cells were used for co-culturing with human embryonic stem cells (ESCs), significantly improving PGCs generation efficiency [200].

On the other hand, research on optimizing cell culture medium was carried out since Niels Geijsen et al. derived PGCs from ESCs by culturing with leukemia inhibitory factor in 2004 [201]. Until now, there have been several biochemical agents used in inducing PGCLCs differentiation in vitro, such as retinoic acid promoting the differentiation of PGCLCs from skin-derived stem cells [202]; retinoic acid combined with CHIR99021 promoting the differentiation of PGCLCs from human ESCs [203]; luteinizing hormone regulating the proliferation of porcine PGCLCs through ceRNA network [204]. Furthermore, recent research has identified the role of epigenetic modification in the differentiation of PGCLCs in vitro. MIR-10B has been discovered to play a role in differentiating PGCLCs from human mesenchymal stem cells [205]. In addition, α-ketoglutarate can promote PGCLCs specialization by regulating epigenetic reprogramming [206]. Similarly, the cell adhesion microenvironment was found to contribute to the differentiation of ESCs, which provide new ideas for PGCs derivation in vitro, where mesh substrates were found to induce self-organize and differentiation of ESCs, transiting to a PGCs-like state without the addition of biochemical inducers [207]. Interestingly, sex differences were found in the associations between Bisphenol A and PGCLC proliferation, with downregulated X-linked gene expression and PGCLC proliferation inhibited in XX cells but not in XY cells [208], providing a theoretical basis for intervening in the fate of different gender PGCs. 

The differentiation of PGCs to embryonic germ cells (EGCs) has a lot of promise in studying the mechanisms of PGC survival, proliferation, and regulation. During the conversion process from PGCs to EGCs, the whole-transcriptome analysis revealed that *BLIMP1* and *Akt* were involved in the specification and reprogramming of PGCs, respectively [209]. Further research showed that *Akt* activation promoted G1-S transition and enhanced PGCs reprogramming by downregulating H3K27me3 [210]. In addition, methylation changes at imprinting control centers (ICCs) during this conversion process were also discovered, stating that methylated ICCs are critical for PGCs derivation from ESCs [211]. Moreover, many new cell models have been developed to study factors regulating PGCs biologies, such as induced pluripotent stem cells [212] and PGCs derivation from nuclear transfer ESCs [213].

### 4.3. Migration of PGCs

Successful migration of PGCs to gonads is essential for gametogenesis in mammal, while anomalous migration of PGCs is required for the origin of endometriosis [214]. Although, with molecular biology development, the stages in PGC migration, with the underlying transcriptional regulatory network and signal pathways, have gradually been discovered and reviewed before [10,215,216,217], how PGCs migrates remains an important question.

Following PGC specification, PGCs first move from the primitive streak to allantois, where members of the interferon-inducing transmembrane protein (*IFITM*) family play a role in PGC incorporation into the hindgut [218]. Subsequently, the hindgut elongated, and PGCs moved into the dorsal mesentery through a fragmented basement membrane and finally colonized the GRs. Jingjing Sun et al. [219] found that, in the absence of *MSX1* and *MSX2*, PGCs migration defected. The number of PGCs was reduced due to the reduction in the expression level *WNT5A**,* which promoted directional migration of PGCs [220]. With improved molecular technology, several other regulatory RNAs have been discovered in the recent years such as *NUP50* [221], *SMAD4* [222], *XVLG1* [223], *HSP70* [224], *PRDM1* [225], *Ptch2*/*Gas1* and *Ptch1*/*Boc* [226]. 

During migration, the epigenome of PGCs undergoes comprehensive remodeling, including global DNA-demethylation, erasure of genomic imprints, and removal of H3K9me2; however, how they occur in PGCs is yet unknown. Anna Mallol et al. identified that *PRDM14* was involved in global and X-chromosomal reprogramming, which upregulated the repressive H3K9me2 dose dependently and removed H3K27me3 from the inactive X-chromosome [227]. In addition, the DNA methylome between human PGCs and mice PGCs was found to be roughly comparable before PGCs differentiation [165], providing a basis for the future establishment of animal models in epigenetic research. However, a recent study indicated that PGCs migration mechanisms vary among mammals. PE Høyer et al. found an association between human PGCs and autonomic nerve fibers, which suggested that PGCs might be guided by nerve fibers [228], which was confirmed by Mollgard K et al. [229]. However, in mice and a non-human primate (marmoset monkey), most PGCs maintained a minimum distance of 50 µm from the closest neuron during different stages of embryonic development. More importantly, PGCs were discovered to reach the gonads before the emergence of neurons around the gonads [230]. Above all, whether PGCs migration mechanisms in different species are diversified remains controversial.

Another factor that affects PGCs migration is DNA damage response (DDR) which is present at all embryonic development stages and results in apoptosis or delayed proliferation of PGCs. However, the underlying mechanisms remain partially known. Recent genetic studies showed that *FANCM* or *MCM9* deficiency reduced the number of PGCs before and after arriving in gonads. Interestingly, *FANCM- MCM9*-double-knockout mice showed an additive reduction of PGCs number [231], indicating that different DDR pathways can cause impaired PGCs migration. In another recent study, conditional knockout of *PRMT5* activated DDR inducing sterility through PIWI-interacting RNA (piRNA) pathway indicated that *PRMT5* was an important DNA protector [232]. The DDR was further studied with Ionizing radiation (IR), where, following germ cell differentiation and uncoupling of meiotic initiation in IR-treated female PGCs, gender differences were observed. In contrast, piRNA metabolism repression and transposon de-repression were detected in IR-treated male PGCs [233]. Importantly, this work provided new ideas for the research on sex manipulation by identifying genes that fit the established XX or XY germline.

### 4.4. Proliferation of PGCs and Gametogenesis 

The PGCs begin to increase during migration and continue until a global change in gene expression occurs; PGCs are ready for gametogenesis. However, the mechanisms regulating the balance between proliferation and differentiation of PGCs remained unclear. Andrea V Cantú et al. discovered that *WNT5A* involves the proliferation of PGCs in different niches by repressing *β-catenin*-dependent and *ROR2*-mediated pathways [234], revealing that the tissue microenvironment regulated PGCs proliferation during migration rather than embryonic age. Another research using conditional knockout models showed that *MASTL* is vital for anaphase entry in female PGCs. Simultaneous deletion of *PPP2R1A* in *MASTL*-knockout PGCs can rescue the failure of PGCs to proceed beyond the metaphase-like stage, demonstrating that *MASTL* with *PPP2A* was essential for establishing female germline by regulating PGCs proliferation through phosphatase activity [235].

Furthermore, proteomic techniques were used to investigate PGC proliferation mechanisms, and it was reported that fatty acid degradation might play an important role in PGC proliferation. Furthermore, in vitro experiments demonstrated that when fatty acid degradation was suppressed, the number of PGCs decreased. Moreover, the expression levels of *AMPK* (*p53* activator to induce cell cycle arrest), phosphorylated *AMPK*, phosphorylated *p53*, and cyclin-dependent kinase inhibitor 1 were increased, indicating that fatty acid degradation is involved in the proliferation of female PGCs via the *p53* pathway [236]. Interestingly, some genes functioned at both proliferation and differentiation. For instance, *ERK1-2* was expressed in PGCs at E8.5–E10.5 and gradually increased from E12.5–E14.5. After culturing PGCs with U0126 (*MEK*-specific inhibitor), *ERK-12* expression was repressed, reducing PGCs at E8.5. Moreover, there were sex differences in controlling meiosis that only progression through meiotic prophase I of female PGCs treated with U0126 were slowed down [237]. In addition to participating in sexual differentiation of bipotential gonad, *FGF9* was dose-dependent in regulating mice XY PGCs fate. Low doses of *FGF9* (0.2 ng/mL) increased male-specific genes expression (*DNMT3L* and *NANOS2*) in XY PGCs, while a high dose of *FGF9* (25 ng/mL) repressed the expression of male-specific genes and stimulated XY PGCs proliferation, revealing that *FGF9* regulates the balance between proliferation and differentiation of XY PGCs in a dose-dependent manner [238]. These could be used as a selective mechanism to favor male or female migrators by repressing the proliferation or differentiation of one through conditional knockouts or conditional overexpression. Interestingly, *EMX2* regulated the *FGF9* pathway in somatic cells [239], which was important for GR formation, demonstrating that sex determination occurs throughout mammals’ lives.

Before gametogenesis, PGCs required permission to start meiosis and sexual differentiation; however, it remained unknown whether this permission was cell-autonomous or gonad-independent. Yueh-Chiang Hu et al. built a *GATA4* (gene only expressed in somatic cells) conditional knockout model. They found that PGCs in *GATA4*-knockout embryos can migrate to the genital ridge but fail to start meiotic [240], indicating that gonad signaling is essential for gametogenesis. To fully explore the function of gonads, single-cell transcriptomics analysis was used in human fetal gonads. Four major signaling pathways (*WNT*, *NOTCH*, *TGFβ*/*BMP*, and receptor tyrosine kinases) were found to be involved in ligand-receptor interactions between PGCs and gonadal somatic cells using the CellPhoneDB algorithm [241]. *WNT* signaling pathway has been studied in depth because it is believed to be involved in sex determination throughout the life cycle. Anne-Amandine Chassot et al. found that spermatogonial proliferation was repressed and spermatocyte apoptosis increased following activation of the *WNT*/*β-catenin* pathway [242], which is consistent with the theory mentioned above that the *WNT* signaling pathway inhibits male-related biological processes. Another study identified *WNT* signaling as a “central gatekeeper” in female gametogenesis. PGCs maintained pluripotency or entered prematurely in the β-catenin gain- and loss-of-function models.

Additionally, by interacting with *POU5F1*, *β-catenin* was involved in pluripotency maintenance, and germ cell differentiation occurred when the *WNT*/*β-catenin* pathway was repressed after *ZNRF3* upregulation [243]. The *FGF* signaling has been shown to regulate PGCs differentiation in two ways, i.e., by repressing female-related gene expression and activating downstream nodal/activin signaling to promote male gamete differentiation through degrading retinoic acid [244] and by activating the expression of *NANOS2* (male germ cell marker) [245], which can prevent XX PGCs meiosis and induce male-like differentiation [246]. Interestingly, Quan Wu et al. found that *SMAD2*, a putative gene downstream from nodal/activin signaling, promoted male differentiation through a retinoic acid-independent routine because retinoic acid signaling suppression did not rescue male-specific gene expression in *SMAD2* conditional knockout testes [247]. In mice PGCs proteomic research, there was no close correlation between proteomic data with published transcriptomic data using comparative analysis [248], revealing that the molecular mechanisms of gametogenesis may extend beyond the scope of the transcriptome, providing us essential inspiration for human gametogenesis research. 

PGCs undergo a wide range of epigenetic reprogramming before sex-specific differentiation. DNA methylation has been extensively studied in PGCs, associated with chromatin reorganization, genomic imprinting erasure, and X-chromosome reactivation [165,249,250]. This process is mainly achieved by repressing DNA methylation-related genes (such as *DNMT3A*/*B*/*L*) and activating *TET* proteins, though there are still many unknowns in this field. Several upstream regulatory genes have been identified using gene-editing technology such as *PRDM14* [251]. Moreover, Peter W S Hill et al. found that *TET1* was involved in the maintenance of DNA demethylation rather than activation, providing a complete understanding of the *TET* family [252]. The *SMARCB1* was discovered to have gender differences in regulating PGC epigenetic reprogramming. In *SMARCB1*-null female mice, meiosis-related genes were repressed, resulting in defects in synapse formation and DNA double-strand break repair. In contrast, in mutant male mice, the expression of genes related to growth and de novo DNA methylation was abnormal, resulting in mitotic arrest delay and hypomethylation of retrotransposons and imprinted genes [253].

Furthermore, *DND1* was identified as a negative regulator of pluripotency and a positive regulator of epigenetic modifiers in male germ cell differentiation. In *DND1*^Ter/Ter^ mutant mice, genes associated with pluripotency, cell cycle, male differentiation, and chromatin regulation were repressed, translating into entering G1/G0 impairment and teratomas formation [254]. These findings supported manipulating sex-dependent differentiation of PGCs; however, the function of these genes in humans remains unknown. As a result, recent research has examined the transcriptome and DNA methylome landscapes of human PGCs, laying the groundwork for understanding the complex relationship between gene regulatory networks and DNA methylation during the global epigenetic reprogramming process of human PGCs [165]. In addition, DNA methylomes of human PGCs during epigenetic reprogramming were roughly similar to mice [165,249]. However, human PGCs also show a unique gene regulatory network in epigenetic modification different from mice PGCs [249,250]. On the other hand, recent studies have identified additional epigenetic reprogramming of PGCs before sexual differentiation, such as histone acetylation [255] and noncoding RNAs [256].

Under the influence of a male or female regulatory network, PGCs give rise to spermatogonial stem cells or oogonia. It is worth noting that the previously mentioned antagonistic network still determines the fate of these germ cells. *WNT4*, activated by *CTNNB1* signaling, can suppress spermatogonial stem cell activity in Sertoli cells [257], while female germ cell survival in the ovary is maintained by the *WNT4*/*β-catenin* pathway [258].

## 5. Conclusions

DSD has been a problem for humans for many years, and identifying functional variants of sex-related genes in DSD patients remains challenging. Infertility is another significant medical issue for which no effective treatments exist. The mechanisms underlying reproductive organogenesis and gametogenesis remain unknown despite considerable progress in recent years. However, it is worth mentioning that we have identified the interaction between somatic and germ cells and that signaling from somatic cells was essential for the proliferation and differentiation of PGCs. In contrast, female germ cells contributed to ovary maintenance. In addition, it was found that genes from males- and female-promoting antagonistic network primarily regulated the mammalian sex determination, which begins during embryonic development and continues throughout the life cycle. Nevertheless, several genes in this antagonistic network are also involved in the biological processes of organ maintenance and development, limiting the application of transgenic technology. In addition, transgenic efficiency remains low, and the sex-reversal trait cannot be stably transmitted to the next generation.

Alternatively, epigenetic changes during reproductive organogenesis and gametogenesis may explain the inability to identify DSD through genetic diagnosis. In order to address the issues above, it is possible to divide further future research into three distinct areas: (i) Intercellular signaling mechanisms must be first investigated, (ii) the structure and regulatory regions of known sex-related genes, intergenic regulatory networks, and identification of novel sex-related genes should be focused on in the future, and (iii) using sequencing technology, changes in DNA methylation, histone modifications, non-coding RNA, and RNA methylation need to be identified during sex determination. With additional research, we will better understand the processes underlying the development of the gonad and germline in humans, mice, and other mammals, which will aid in diagnosing and treating DSD and human infertility. In addition, these studies can offer theoretical support for manipulating offspring sex ratios in livestock production.

## Figures and Tables

**Figure 1 ijms-23-07500-f001:**
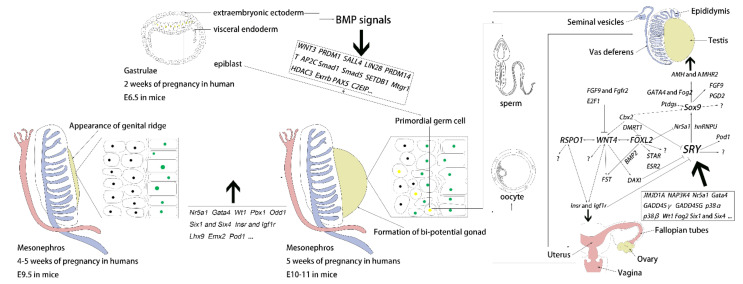
Schematic illustrations of the development of gonad and primordial germ cells (PGCs). At 4–5 weeks of pregnancy in humans (E9.5 in mice), coelomic epithelial cells (green) start to proliferate on the ventromedial surface of the mesonephros, and the genital ridge (brown) appears. At five weeks of pregnancy in humans (E10–E11 in mice), coelomic epithelial cells continue to proliferate and form bipotential gonads under the regulation of *NR5A1*, *GATA4*, etc. Subsequently, bipotential gonad differentiates into testis and ovary, respectively, through a sex-related genes antagonistic network. Among them, *SRY*, *WNT4*, *RSPO1*, and *FOXL2* factors. In addition, the Wolffian duct (blue) and Müllerian duct (pink) form epididymis, vas deferens, and seminal vesicles or fallopian tubes, uterus, and part of the vagina, respectively. On the other hand, PGCs originate from a subpopulation of cells in the proximal epiblast (yellow). At two weeks of pregnancy in humans (E6.5 in mice), extraembryonic ectoderm and visceral endoderm secrete signals BMP signals, in turn, activate the expression of *WNT3*, *PRDM1*, etc. and induce the specification of PGCs. At five weeks of pregnancy in humans (E10–E11 in mice), PGCs have migrated into the bipotential gonad. After several rounds of cell division and a global change in gene expression, PGCs differentiate into sperm or oocyte after receiving signals from the forming testis or ovary and the nearby mesonephric tissue.

**Table 1 ijms-23-07500-t001:** Genes involved in genital ridge formation.

Genes	Function in Organogenesis and DSD	References
*NR5A1*	*NR5A1* is involved in the development of the gonad, adrenal gland, and pituitary.	[25,26,27]
	*NR5A1* variants are associated with male infertility and DSD in humans males and females.	[28,29,30,31,32,33,34]
*GATA4*	*GATA4* is required for the proliferation of coelomic epithelial cells and is involved in gonadal development by interacting with *FOG2*.	[35,36]
	The *GATA4* mutant protein failed to bind with *F**OG2*, resulting in DSD.	[37]
*WT1*	*WT1* plays a distinct role in gonadal formation and development by maintaining somatic cell survival.	[38]
	*WT1* mutations are responsible for Frasier syndrome with streak gonads.	[39,40]
*LHX9*	*LHX9* participates in gonad formation by regulating cell proliferation.	[41]
*EMX2*	*EMX2* is required for the migration and survival of cells in the mesenchymal compartment and involves GR formation by regulating *NR5A1* expression.	[42]
*SIX1* and *SIX4*	*SIX1* and *SIX4* have a functional redundancy and mainly function in the proliferation of supporting cell precursors and steroidogenic cell precursors.	[43]
*POD1*	*POD1* is essential for gonadal development by regulating *Nr5a1* expression.	[44]
*CBX2*	*CBX2* is required for splenic vascular, adrenal gland, and gonad formation.	[45,46]
*INSR* and *IGF1R*	*INSR* and *IGFIR* regulate somatic progenitor cell proliferation by mediating Insulin and its growth factors (IGF1 and IGF2) during GR formation.	[47]
*PBX1*	*PBX1* is involved in progenitor cell proliferation in GR by regulating the expression of *NR5A1*.	[48]
	*PBX1* mutation abolishes its interaction with *CBX2* and *EMX2*, causing gonadal dysgenesis and radiocubital synostosis in humans.	[49]
*ODD1*	*ODD1* regulates the expression of *LHX1*, *PAX2*, and *WT1*, inhibiting cell apoptosis in nephrogenic mesenchyme and participating in gonadal development.	[50]

**Table 2 ijms-23-07500-t002:** Genes involved in gonadal sexual differentiation.

Genes	Functions	References
*JMJD1A*	*JMJD1A* is involved in the H3K9 demethylation in *SRY*, and *JMJD1A* deficiency presents a decrease in *SRY* expression.	[135]
*MAP3K4*	*MAP3K4* is involved in *SRY* expression, and loss of *MAP3K4* will lead to a male sex reversal.	[136]
*GADD45γ*	*GADD45γ* is upstream of *MAP3K4*, without which will cause male sexual reversal.	[72]
*p38α* and *p38β*	*p38α* and *p38β* are members of *p38 MAPK* family, and loss of *p38α* and *p38β* will causes disruption to *SRY* expression and XY embryonic gonadal sex reversal.	
*GADD45G*	*GADD45G* is necessary for *SRY* expression.	[137]
*HNRNPU*	*HNRNPU* enhances the expression of *SOX8* and *SOX9* by interacting with *WT1* and *SOX9*.	[138]
*AMH* and *AMHR2*	*AMH* and its receptor *AMHR2* are involved in the normal development of the accessory gland.	[139]
*NR5A1*	*NR5A1* is involved in testis formation by cooperating with other regulators such as *WT1*, *DAX1*, *SRY*, and *SOX9*.	[140]
*GATA4*	The interaction of *GATA4* and *FOG2* is important in sex differentiation because it regulates the expression of *SRY* and *AMH*.	[36,141]
*FOG2*		
*PTGDS*	*PTGDS* is one of the downstream targets of *SOX9*, which involves the production of prostaglandin D2, maintaining the sustained expression of *SOX9* in testis.	[142,143]
*WT1*	*WT1* is a potential upstream of *SRY* and controls somatic cells’ fate through regulating *NR5A1* expression.	[38,144]
	*WT1* variants lead to 46,XX testicular DSD in humans.	[145,146]
*SIX1* and *SIX4*	*SIX1* and *SIX4* regulate *SRY* expression by activating *FOG2* expression, regulating male sex determination.	[43]
*POD1*	The downstream target of *SRY* is *POD1*, which is involved in the formation of testicular cords and testis-specific coelomic vessels.	[44,147]
*CBX2*	*CBX2* regulates *SRY* expression by interfering with upstream steps.	[46,148]
*INSR* and *IGF1R*	*INSR* and *IGF1R* have potential feedback interactions between *WNT4* and *RSPO1* signaling pathways.	[84]
	*INSR* and *IGF1R* are involved in the adrenal specification, testicular differentiation, and ovarian development by regulating the expression of sex-related genes, including *WT1*, *LHX9*, and *NR5A1*.	[149]
*FGF9* and *PGD2*	*FGF9* and *PGD2* are downstream of *SOX9* and are involved in supporting-to-Sertoli cell differentiation by activating testis-related genes and repressing anti-testis genes.	[79]
*E2F1*	*E2F1* regulates testicular descent and controls spermatogenesis by repressing *WNT4* expression	[150]
*FGF9* and *FGFR2*	*FGF9* and *FGFR2* are required in testis development by repressing the expression of *WNT4* and *FOXL2*.	[151,152]
*FST*	*FST* prevents testis-specific vasculature formation by antagonizing Activin B action through *WNT4* activation.	[153]
*DAX1*	*DAX1* is downstream of *WNT4* and is involved in gonadal sexual differentiation by antagonizing *SRY* expression.	[154]
*BMP2*	*BMP2* acts cooperatively with *FOXL2* to regulate *FST* gene expression during ovarian development.	[155]
*DMRT1*	*DMRT1* is required for sexual differentiation of somatic and germ cells by silencing *FOXL2* expression.	[156]

**Table 3 ijms-23-07500-t003:** Positive and negative signals directed PGC fate.

Genes	Functions	References
*WNT3*	*WNT3* enables PE cells to receive a BMP4 signal.	[168]
*PRDM1*	*PRDM1* is Likely downstream of *BMP4*.	[169]
	*PRDM1* is involved in PGCs formation by repressing somatic cell program genes through selective recruitment of *HDAC3*.	[170]
*SALL4*	*SALL4* participates in the specification of PGCs by suppressing the expression of somatic cell program genes.	[171]
*LIN28*	*LIN28* is involved in PGCs development by regulating *PRDM1* transcript translation.	[172]
*PRDM14*	*PRDM14* establishes germ cell lineage by inducing *SOX2* expression and cooperating with *TFAP2C* and *PRDM1*, which upregulates pluripotency genes and represses somatic markers.	[173,174]
*T*	*T* specifies germ cell fate by activating the expression of *PRDM1* and *PRDM14*.	[175]
*AP2C*	*AP2C* is most likely downstream of *PRDM1* and is involved in maintaining PGCs.	[176]
*SMAD1*	*SMAD1* is a downstream signal mediator for BMPs and is essential for PGCs formation.	[177]
*SMAD5*	*SMAD5* is a downstream signal mediator for BMPs and is required for PGCs development.	[178]
*SETDB1*	*SETDB1* is involved in PGCs fate determination by ensuring BMP signaling through repressing the expression of *Dppa2*, *Otx2*, and *Utf1*.	[179]
*MTGR1*	*MTGR1* is involved in stem cell maintenance and PGCs formation by mediating *PRDM14* functions.	[180]
*ESRRB*	*ESRRB* functions as an upstream factor of *BMP4* and regulates PGCs development.	[181]
*PAX5*	*PAX5* participates in PGCs specification by activating germline and repressing somatic program genes through a *PAX5**-**OCT4**-**PRDM1* core transcriptional network.	[182]
*C2EIP*	*C2EIP* promotes the generation of PGCs by activating the Hedgehog (Hh) signaling pathway via *PTCH2* ubiquitination.	[183]

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
