# Peer review of "Early Gonadal Development and Sex Determination in Mammal"

_ijms, 2022, doi:10.3390/ijms23147500_

Round 1

Reviewer 1 Report

The manuscript is interesting but very descriptive. The reading is a bit tedious. It could be interesting to rework the shape to give it more dynamism. Readability must be improved. I recommend major revision.

MAJOR COMMENT

It would be interesting to highlight more the interest of this review compared to other recent reviews already existing on the subject. How does this narrative review make a significant contribution to the existing literature?

MINOR COMMENTS

Line 18: « entire » seems a bit ambitious

Line 51: mice « ) » The parenthesis is missing

Table 1: NR5A1 not “involves” but “is involved in”?

Line 95: is it 46 XY DSD patients or 46 XX DSD patients?

Table 2

·JMJD1A : not “involve” but “is involved in”?

·“p38α and p38β involve in the pathway that MAP3K4 and GADD45γ regulate SRY expression.” Unclear, please reformulate.

·AMH and its receptor AMHR2 involve in the normal development of the accessory gland: not “involve” but “are involved in”?

·NR5A1 involves: not “involves” but “is involved in”?

·INSR and IGF1R involve in: not “involve” but “are involved in”?

·FGF9 and PGD2 are downstream of Sox9 and involve: “are involved in”?

·DAX1 is downstream of WNT4 and involves: “is involved in”?

Same comments in Table 3.

Line 253: “could provide a theoretical basis for manipulating offspring sex ratios in livestock production by changing the ratio of Y-and X-chromosome-bearing sperm”. How could this be possible? How could gametes not be 50% carriers of X and 50% of Y? Explain further.

Line 260: BMPs is misplaced in the sentence

Lines 283-284: in which species? (Ref 192 and 193)

Author Response

Comments and Suggestions for Authors

The manuscript is interesting but very descriptive. The reading is a bit tedious. It could be interesting to rework the shape to give it more dynamism. Readability must be improved. I recommend major revision.
Response: We appreciate reviewer s' comments and have made a lot of revisions to the article to make it easily read.

MAJOR COMMENT

It would be interesting to highlight more the interest of this review compared to other recent reviews already existing on the subject. How does this narrative review make a significant contribution to the existing literature?

Response: Thank you for the constructive advice. Although some studies in this field have been reviewed elsewhere, we combined single-cell RNA sequencing results (Ref 16, 17, 166, 168 and 243) on the basis of these reviews to discuss some potential mechanistic link between bipotential gonads and PGCs and find genes that play key roles in multiple stages of early gonadal development and sex determination [such as GATA4 is involved in genital ridge formation (Ref 29-31), sex differentiation of bipotential gonads (Ref 140 and 141) and PGCs migration (Ref 242); FGF9 is involved in sex differentiation of bipotential gonads (Ref 73) and PGCs (Ref 240); WNT/β-catenin signaling is involved in sex determination throughout the life cycle (Ref 83, 84, 236, 243, 244, 260and 261)]. On the other hand, we described novel gene editing experiments which lead to sex reversal [such as knockout of the HMG domain of the SRY gene leads to boar sex reversal (Ref 60), loss of MAP3K4 kinase activity results in the mice embryonic male gonadal sex reversal (Ref 134)], potential gender differences before sex-specific differentiation[such as gender differences of X-linked genes expression and PGCLCs proliferation after treated with Bisphenol A (Ref 209); gender differences of germ cell differentiation, meiotic, piRNA metabolism and transposon derepression after IR-treatment during PGCs migration (Ref 235); gender differences of progression through meiotic prophase I of PGCs treated with U0126 (MEK-specific inhibitor) (Ref 239); SMARCB1 was discovered to have gender differences in regulating PGC epigenetic reprogramming (Ref 256)], and the function of epigenetic regulation in these processes [such as DNA methylation, histone modifications and Non-coding RNAs in sex determination of bipotential gonads (Ref 132); LncPGCAT-1 can positively regulate the formation of PGCs by elevating the expression of Cvh and C-kit and repressing the NANOG in vitro and in vivo (Ref 167); MIR-10B plays a role in differentiating PGCLCs from human mesenchymal stem cells (Ref 206); DNA-demethylation during PGCs migration (Ref 229), function of piRNA in DNA damage response during PGCs migration (Ref 234); epigenetic reprogramming before sex-specific differentiation of PGCs (Ref 251-253)]. Therefore, based on the comments, we have re-described the introduction and described how this review contribute to the existing literature in revised manuscript (line 39).

MINOR COMMENTS

Line 18: « entire » seems a bit ambitious

Response: We appreciate the reviewer’s suggestion. We have deleted ‘entire’ in revised manuscript (line 18).

Line 51: mice « ) » The parenthesis is missing

Response: Corrected accordingly. Thanks for the suggestion.

Table 1: NR5A1 not “involves” but “is involved in”?

Response: Corrected accordingly. Thanks for the suggestion.

Line 95: is it 46 XY DSD patients or 46 XX DSD patients?

Response: Thank you for the constructive advice. It is 46,XX DSD patients and We have corrected it in revised manuscript (line 100).

Table 2

·JMJD1A : not “involve” but “is involved in”?

Response: Corrected accordingly. Thanks for the suggestion.

·“p38α and p38β involve in the pathway that MAP3K4 and GADD45γ regulate SRY expression.” Unclear, please reformulate.

Response: Thanks for the suggestion. We have rephrased our descriptions in revised manuscript (Table 2).

·AMH and its receptor AMHR2 involve in the normal development of the accessory gland: not “involve” but “are involved in”?

Response: Corrected accordingly. Thanks for the suggestion.

·NR5A1 involves: not “involves” but “is involved in”?

Response: Corrected accordingly. Thanks for the suggestion.

·INSR and IGF1R involve in: not “involve” but “are involved in”?

Response: Corrected accordingly. Thanks for the suggestion.

·FGF9 and PGD2 are downstream of Sox9 and involve: “are involved in”?

Response: Corrected accordingly. Thanks for the suggestion.

·DAX1 is downstream of WNT4 and involves: “is involved in”?

Response: Corrected accordingly. Thanks for the suggestion.

Same comments in Table 3.

Response: Corrected accordingly. Thanks for the suggestion.

Line 253: “could provide a theoretical basis for manipulating offspring sex ratios in livestock production by changing the ratio of Y-and X-chromosome-bearing sperm”. How could this be possible? How could gametes not be 50% carriers of X and 50% of Y? Explain further.
Response: We appreciate the reviewer’s suggestion. Due to the potential gender differences during the migration and differentiation of PGCs, gene editing can be used to control gamete sex and we have rephrased our descriptions in revised manuscript (line 255).

Line 260: BMPs is misplaced in the sentence

Response: Corrected accordingly. Thanks for the suggestion.

Lines 283-284: in which species? (Ref 192 and 193)

Response: Thanks for the suggestion. Ref 192 and 193 are in mice and we have added this content in revised manuscript (line 286).

Reviewer 2 Report

The manuscript entitled "Early Gonadal Development and Sex Determination in Mammal" by Xie et al. is a well-prepared material collecting and summarizing the knowledge to date on the molecular regulation of the development of gonads at the earliest stage, development of the genital ridges, sex determination, differentiation, and migration of the primordial germ cells. The text is very well organized and clearly written.

However, I found many editing errors that need to be corrected. All gene symbols must be in italics, both in the text and in tables and figures. I am enclosing PDF with my comments.

For several years, no article has been published that would collect the latest research results on the gonad development and sex determination, thus an update was needed. So, I highly recommend publishing this article after minor corrections. 

Author Response

Comments and Suggestions for Authors

The manuscript entitled "Early Gonadal Development and Sex Determination in Mammal" by Xie et al. is a well-prepared material collecting and summarizing the knowledge to date on the molecular regulation of the development of gonads at the earliest stage, development of the genital ridges, sex determination, differentiation, and migration of the primordial germ cells. The text is very well organized and clearly written.

However, I found many editing errors that need to be corrected. All gene symbols must be in italics, both in the text and in tables and figures. I am enclosing PDF with my comments.

For several years, no article has been published that would collect the latest research results on the gonad development and sex determination, thus an update was needed. So, I highly recommend publishing this article after minor corrections. 

Response: Thank you so much for your kind comment and we have improved the manuscript accordingly.

Line 29: a citation needed.

Response: We agree with the Reviewer that citation for the percent of couples affected by infertility are necessary and we have added it in the revised manuscript (line 29).

Line 32: in most or in all?

Response: Thank you for the constructive advice. Sex determination is governed by sex chromosomes in all mammals and have modified the manuscript (line 31).

Line 33: “bi-potential” should be changed to “bipotential”.

Response: Corrected accordingly. Thanks for the suggestion.

Line 51: mice « ) » a bracket is missing here.

Response: Corrected accordingly. Thanks for the suggestion.

Line 52: “bi-potential” should be changed to “bipotential”.

Response: Corrected accordingly. Thanks for the suggestion.

Line 59: All gene symbols must be in italics.

Response: Corrected accordingly. Thanks for the suggestion.

Table 1: All gene symbols must be in italics.

Response: Corrected accordingly. Thanks for the suggestion.

Figure 1: All gene symbols must be in italics.

Response: Corrected accordingly. Thanks for the suggestion.

Line 64: The authors type "E 9.5" and E9.5". It must be unified throughout the text.

Response: We thank the reviewer so much for pointing out this issue and giving us a chance to correct our negligence. We have modified the type and they were used uniformly in the revised manuscript (line 69).

Line 67: All gene symbols must be in italics. “bi-potential” should be changed to “bipotential”.

Response: Corrected accordingly. Thanks for the suggestion.

Line 69: All gene symbols must be in italics.

Response: Corrected accordingly. Thanks for the suggestion.

Line 74: All gene symbols must be in italics.

Response: Corrected accordingly. Thanks for the suggestion.

Line 79: “bi-potential” should be changed to “bipotential”.

Response: Corrected accordingly. Thanks for the suggestion.

Line 81: “bi-potential” should be changed to “bipotential”.

Response: Corrected accordingly. Thanks for the suggestion.

Line 86: All gene symbols must be in italics.

Response: Corrected accordingly. Thanks for the suggestion.

Line 87: All gene symbols must be in italics.

Response: Corrected accordingly. Thanks for the suggestion.

Line 95: “46 XY” should be changed to “46,XY”.

Response: Corrected accordingly. Thanks for the suggestion.

Line 113: The authors type "E 10.5" and E10.5". It must be unified throughout the text.

Response: We thank the reviewer so much for pointing out this issue and giving us a chance to correct our negligence. We have modified the type and they were used uniformly in the revised manuscript (line 119).

Line 132: “46 XX” should be changed to “46,XX”.

Response: Corrected accordingly. Thanks for the suggestion.

Lin 152: the abbreviation should be developed here: (SEx Reversion, Kidneys, Adrenal and Lung dysgenesis).

Response: Thank you for the suggestion and we have developed the abbreviation in the revised manuscript (line 156).

Line 160: “46 XX” should be changed to “46,XX”.

Response: Corrected accordingly. Thanks for the suggestion.

Line 168: All gene symbols must be in italics.

Response: Corrected accordingly. Thanks for the suggestion.

Line 187: “46, XY” should be changed to “46,XY”.

Response: Corrected accordingly. Thanks for the suggestion.

Line 190: It is a gene symbol so it should be italicized.

Response: Corrected accordingly. Thanks for the suggestion.

Table 2: All gene symbols must be in italics.

Response: Corrected accordingly. Thanks for the suggestion.

Table 3: All gene symbols must be in italics.

Response: Corrected accordingly. Thanks for the suggestion.

Reviewer 3 Report

The review titled "Early Gonadal Development and Sex Determination in Mammal" is described clearly and in detail. It addresses current issues and is supported by scientific studies.

I believe the review can be published because of the importance of the topic and the clarity of the content. 

Author Response

Comments and Suggestions for Authors

The review titled "Early Gonadal Development and Sex Determination in Mammal" is described clearly and in detail. It addresses current issues and is supported by scientific studies.

I believe the review can be published because of the importance of the topic and the clarity of the content.

Response: Thank you for your comment. We are glad the Reviewer has found our article suitable for publication.

Round 2

Reviewer 1 Report

This revised version of the manuscript meets the standards of IJMS.